# Differential Responses of Colorectal Cancer Cell Lines to *Enterococcus faecalis’* Strains Isolated from Healthy Donors and Colorectal Cancer Patients

**DOI:** 10.3390/jcm8030388

**Published:** 2019-03-20

**Authors:** Carolina Vieira De Almeida, Matteo Lulli, Vincenzo di Pilato, Nicola Schiavone, Edda Russo, Giulia Nannini, Simone Baldi, Rossella Borrelli, Gianluca Bartolucci, Marta Menicatti, Antonio Taddei, Maria Novella Ringressi, Elena Niccolai, Domenico Prisco, Gian Maria Rossolini, Amedeo Amedei

**Affiliations:** 1Department of Surgery and Translational Medicine, University of Florence, 50134 Florence, Italy; almeida.cv@gmail.com (C.V.D.A.); antonio.taddei@unifi.it (A.T.); marianovella.ringressi@unifi.it (M.N.R.); 2Department of Experimental and Clinical Biomedical Sciences “Mario Serio”, University of Florence, 50134 Florence, Italy; matteo.lulli@unifi.it (M.L.); nicola.schiavone@unifi.it (N.S.); 3Department of Experimental and Clinical Medicine, University of Florence, 50134 Florence, Italy; vincenzo.dipilato@unifi.it (V.d.P.); edda.russo@unifi.it (E.R.); giulia.nannini@unifi.it (G.N.); simone.baldi1@stud.unifi.it (S.B.); rossella.borrelli92@gmail.com (R.B.); elena.niccolai@unifi.it (E.N.); domenico.prisco@unifi.it (D.P.); gianmaria.rossolini@unifi.it (G.M.R.); 4Department of Neurosciences, Psychology, Drug Research and Child Health Section of Pharmaceutical and Nutraceutical Sciences University of Florence, 50139 Florence, Italy; gianluca.bartolucci@unifi.it (G.B.); marta.menicatti@unifi.it (M.M.); 5Department of Microbiology and Virology Unit, Florence Careggi University Hospital, 50134 Florence, Italy; 6Department of Biomedicine, Azienda Ospedaliera Universitaria Careggi (AOUC), 50134 Florence, Italy

**Keywords:** colorectal cancer, *Enterococcus faecalis*, bacterial metabolites, gut microbiota, tumor cell lines

## Abstract

The metabolites produced by the host’s gut microbiota have an important role in the maintenance of intestinal homeostasis, but can also act as toxins and induce DNA damage in colorectal epithelial cells increasing the colorectal cancer (CRC) chance. In this scenario, the impact of some of the components of the natural human gastrointestinal microbiota, such as *Enterococcus faecalis* (*E. faecalis*), at the onset of CRC progression remains controversial. Since under dysbiotic conditions it could turn into a pathogen, the aim of this study was to compare the effect of *E. faecalis*’ strains (isolated from CRC patients and healthy subjects’ stools) on the proliferation of different colorectal cells lines. First, we isolated and genotyping characterized the *Enterococcus faecalis*’ strains. Then, we analyzed the proliferation index (by 3-(4,5-Dimethylthiazol-2-yl)-2,5-Diphenyltetrazolium Bromide (MTT) assay) of three tumor and one normal intestinal cell lines, previously exposed to *E. faecalis* strains pre-cultured medium. Stool samples of CRC patients demonstrated a reduced frequency of *E. faecalis* compared to healthy subjects. In addition, the secreted metabolites of *E. faecalis*’ strains, isolated from healthy donors, decreased the human ileocecal adenocarcinoma cell line HCT-8 and human colon carcinoma cell line HCT-116 cell proliferation without effects on human colorectal adenocarcinoma cell line SW620 and on normal human diploid cell line CLR-1790. Notably, the metabolites of the strains isolated from CRC patients did not influence the cell growth of CRC cell lines. Our results demonstrated a new point of view in the investigation of *E. faecalis*’ role in CRC development, which raises awareness of the importance of not only associating the presence/absence of a unique microorganism, but also in defining the specific characteristics of the different investigated strains.

## 1. Introduction

Colorectal cancer (CRC) is one of the most commonly diagnosed cancers among both men and women worldwide, being the third most frequent in many high-income countries, with an estimated more than 100,000 new cases expected in 2018 [1]. The CRC incidence in low-income countries is closely related with differences in lifestyle [2,3], with only 15% of cases having a familial feature, whereas sporadic forms represent 85% [4]. Environmental factors, such as smoking, alcoholism, obesity, sedentary lifestyle, consumption of red meat, high-fat diet, and inadequate fiber intake, are closely involved in the CRC onset and progression [5]. All these risk factors also have a modulating role on the host gut microbiota (GM) composition, whose effects on CRC progression/protection have been investigated during the last years [6,7]. The gut microbiota, a natural defensive barrier to infections, is involved in several physiological functions and plays a key role in maintaining the gut homeostasis [8]. The GM members can modulate the mucosal immune system, as well as directly change the expression of some host genes associated with nutrient uptake, metabolism, angiogenesis, and mucosal barrier functions [9,10].

The GM’s protective role against intestinal diseases is closely linked with its ability to ferment a range of dietary substances that are not completely digested and absorbed in the small intestine. The microbial carbohydrate fermentation, for example, produces short chain fatty acids (SCFAs), such as acetic, propionic, and butyric, which can be further metabolized by mammalian cells for energy, thus having beneficial effects. On the other hand, bacterial transformation of dietary components and other chemicals in the intestinal lumen is associated with the production of carcinogenic agents, whose damaging effects on colonic mucosal cells can influence cancer development [11]. In fact, approximately 20% of cancers are associated with microbes [10], especially CRC [12,13,14], where a dynamic crosstalk exists between intestinal epithelial cells, the microbes (that colonize their apical surface), and the surrounding local immune cells [15], which, in turn, have a key role in CRC progression, especially T cells [16].

While some GM members (*Streptococcus bovis*, *Bacteroides*, *Clostridia*, and *Helicobacter pylori*) have an evident role on cancer promotion [17] and others play protective roles (*Lactobacillus* spp. and *Bifidobacterium* spp.) [18], for the symbiotic lactic acid bacteria (LAB), *Enterococcus faecalis* (*E. faecalis*), a controversial role in CRC has been hypothesized. This *Firmicutes* phylum member belonging to the *Enterococcaceae* family, even if used as a probiotic product [19,20] due to its great ability to confer beneficial effects on human health by their fermented products [21], has also been regarded as being possibly involved in CRC development, given its ability to damage the deoxyribonucleic acid (DNA) of colonic epithelial cells [22]. Moreover, previous studies have demonstrated that the feces of CRC patients showed increased concentrations of *E. faecalis*, strengthening its role as cancer promoter [23,24].

Among intestinal *Enterococci*, *E. faecalis* is the most prevalent cultured species found in human feces (10^5^ to 10^7^ Colony Forming Unit (CFU)/g), followed by *E. faecium* (10^4^ to 10^5^ CFU/g), but these proportions change with the host’s geographical location and especially with diet [25]. *E. faecalis* is one of the first colonizers of the human gastrointestinal tract and it has a major impact on intestinal immune development in the very early life stages [26]. In newborns, it plays a protective role, regulating the colonic homeostasis during development by suppressing pathogen-mediated inflammatory responses in human intestinal epithelial cells, inducing interleukin (IL)-10 expression [27] and attenuating proinflammatory cytokine secretions, especially IL-8 [28].

Aiming to verify whether *E. faecalis* strains isolated from CRC patients’ feces have substantial differences with those isolated from healthy donors’ samples, we recruited nine CRC Italian patients and nine healthy donors matched by age. Our results pointed out significant genotyping differences among all the *E. faecalis* strains, mainly regarding the genes of adhesion and virulence factors. Consequently, we analyzed the effect of the metabolites of the different isolated strains on three CRC cell lines’ proliferation (and a normal colon cell line), demonstrating an antiproliferative role only for those strains isolated from healthy donors.

## 2. Material and Methods

### 2.1. Ethical Statement

The Local Ethics Committee (Prot. 2010/0012462) approved the study. Institutional Ethics Committee Statement: All procedures involving the cells of healthy donors were done according to the Declaration of Helsinki and approved by the local Ethics Committee by the AOUC Careggi Institutional Review Board (Prot. 2010/0012462).

### 2.2. CRC Patients and Healthy Donors

Overall, nine patients with CRC and nine healthy donors were enrolled at Careggi University Hospital (Florence, Italy) (Table 1) between April and May 2016. Exclusion criteria included antibiotic intake and use of probiotics/culture milk 2 months before and within the study. All the participants provided an informed written consent prior to enrolment, in compliance with national legislation and the Code of Ethical Principles for Medical Research Involving Human Subjects of the World Medical Association (Declaration of Helsinki).

### 2.3. Isolation and Genotyping of Enterococcus faecalis

To isolate the *E. faecalis* strains, we resuspended 0.25 g of feces in 250 µL of sterile saline (NaCl 0.9%). From this solution, we plated 50 µL onto Columbia CNA (colistin, nalidixic acid) blood agar with crystal violet (CV) (CNA-CV Agar) with 5% sheep blood (Sh) enrichment (CNA-CVSh) (Becton, Dickinson and Company, Franklin Lakes, NJ, USA) and incubated it at 37 °C for 30 h. Then, we isolated single colonies (±0.032 CFU/mg of stool) and cultivated them onto Columbia with 5%sheep blood (COS) medium (Biomerieux, Grassina, FI, Italy) for 24 h at 37 °C. To identify the species’ level of putative *Enterococci* colonies, we performed a Matrix Assisted Laser Desorption/Ionization Time of Flight MALDI-TOF assay (Vitek MS, BioMérieux Inc., Marcy l’Etoile, France). The confirmed *E. faecalis* colonies were stored at −80 °C in brain heart infusion broth medium (BHI) (Oxoid, Altrincham, Cheshire, UK) with 10% glycerol added. When necessary, we grew these strains on COS medium at 37 °C for 24 h before performing different experiments.

To evaluate the clonal diversity of the isolated *E. faecalis* colonies (*n* = 16), we performed a Random Amplified Polymorphic DNA- Polymerase Chain Reaction (RAPD)-PCR assay, as previously described by Martin et al. 2005 [29]. RAPD-PCR was performed on cell lysates obtained by resuspension of single bacterial colonies in 300 µL of Tris- Ethylenediaminetetraacetic acid (TE) buffer, incubation for 15 min 95 °C, and clarification through centrifugation for 5 min at 13,000 rpm.

Quantitative detection of *E. faecalis* was performed by q-PCR on total DNA extracted from subjects’ feces using the DNeasy Power Lyzer Power Soil DNA isolation Kit (MoBio-QIAgen, Valencia, CA, USA). We used a species-specific primer set for the detection of *E. faecalis* and universal primers for the detection of the total bacterial loads to target the 16S rRNA gene, as previously described by Sedgley et al. [30]. The PCR thermal conditions were: 3 min for the initial enzyme activation/DNA denaturing step at 95 °C followed by 44 consecutive cycles at 95 °C for 20 s; (57 °C E16S) (53 °C for U16S) for 45 s; 60 °C for 5 s.

As previously described, whole genome sequencing was performed [31] on selected *E. faecalis* to include all the different clonal profiles, identified according to RAPD-PCR results (*n* = 8). We investigated the clonal relatedness by the determination of: (i) The Multilocus sequence typing (MLST) profile through the MLST 1.8 tool [32] using the assembled whole genome sequences as the input; (ii) the core genome SNP phylogeny, through the CSI Phylogeny 1.4 tool [33], using default parameters and the raw sequence reads as the input. We generated the phylogenetic trees, using the *E. faecalis* ATCC^®^ 29212™ genome (GenBank acc. no. NZ_CP008816.1) as a reference. Identification of prophage sequences was carried out using PHAge Search Tool Enhanced Release (PHASTER) [34]. The presence of virulence factors and antibiotic resistance genes was assessed using the VirulenceFinder 1.5 and the ResFinder 3.0 tools [35], respectively. We performed the sequence comparison by using the Basic Local Alignment Search Tool (BLASTN) software and the nr or wgs databases [36]. Draft genomes of sequenced strains were deposited at National Center for Biotechnology Information (NCBI) as Whole Genome Sequence WGS projects (accession numbers to be assigned).

### 2.4. Secreted Metabolites

To obtain pre-fermented medium, we used the protocol previously described by Grootaert et al. [37] with some modifications. Briefly, the *E. faecalis* strains were cultivated in 5 mL of BHI medium at 37 °C. After 10 h, an equal number of cells (3 × 10^8^ CFU/mL) was centrifuged (5 min, 13,000 rpm) and washed with 1 mL of PBS. The pellets were suspended in 5 mL of Roswell Park Memorial Institute (RPMI) 1640 medium (Thermo Fisher Scientific Inc., Waltham, MA, USA) without antibiotics or fetal bovine serum (FBS) and incubated at 37 °C for 3 h. Thereafter, the suspension was centrifuged (5 min, 4000 rcf) and the supernatant was sterilized with a 0.22 mm filter (Millipore, Billerica, MA, USA) and retained as the pre-fermented medium. We used the commercial ATCC29212 strain as a control (ATCC, Manassas, VA, USA).

### 2.5. Gas Chromatography Mass Spectrometry (GC-MS) Analysis of SCFAs

Methanol and tert-butyl methyl ether (Chromasolv grade), sodium bicarbonate and hydrochloric acid (reagent grade), (2H_3_)Acetic, (2H_3_)Propionic, (2H_7_)iso-Butyric and (2H_9_)iso-Valeric (used as internal standards (ISTDs)), acetic acid, propionic acid, butyric acid, isobutyric acid, valeric acid, and isovaleric acid (analytical standards grade) were purchased by Sigma-Aldrich (Milan, Italy). MilliQ water 18 MΩ was obtained from Millipore’s Simplicity system (Milan, Italy). The SCFAs’ analysis was performed by an Agilent GC-MS system composed with a 5971 single quadrupole mass spectrometer, 5890 gas-chromatograph, and 7673 autosampler.

The SCFAs in the samples (Appendix A) were analyzed as free acid form using a SupelcoNukol column, with a 30 m length, 0.25 mm internal diameter, and 0.25 µm of film thickness with the temperatures program as follows: Initial temperature of 40 °C was held for 1 min, then it was increased to 150 °C at 30 °C/min, and finally increased to 220 °C at 20 °C/min. 1 µL aliquot of the extracted sample was injected in splitless mode (splitless time 1 min) at 250 °C, while the transfer line temperature was 280 °C. The carrier flow rate was maintained at 1 mL/min.

### 2.6. Cell Lines

The used cell lines were purchased from the American Type Culture Collection (ATCC, Manassas, VA, USA). Since primary and metastatic tumor cells have different metabolic, genetic, epigenetic, and morphological characteristics, we decided that it was more appropriate to evaluate the effect of *E. faecalis* metabolites in different CRC cell lines. In detail, we chose HCT-116 and HCT-8 as the primary tumors’ model and SW-620 as a lymphonode metastasis model. The embryo colonic cell line, CLR-1790, was used as the non-tumor control. The culture conditions of each cell line are reported in Table 2.

### 2.7. Treatment of Cell Lines with Pre-Fermented Medium and MTT Assay

The human HCT-116, HCT-8, and SW-620 colon cancer cells, and the normal intestinal CLR-1790 cells were plated on flat-bottomed 96-well culture plates (5 × 10^3^ cells/well) and incubated with their respective culture medium for 24 h at 37 °C under 5% CO_2_ tension. After, we added the pre-fermented medium in a proportion of 1:5 (pre-fermented medium:culture medium). We used RPMI as the control. The effect of the *E. faecalis* secreted metabolites on cell proliferation was determined using the MTT assay [38] after 72 h of culture. Absorbance was measured at 595 nm using the iMark microplate reader (Biorad, Hercules, CA, USA) for the cell viability calculation while the absorbance of the controls was set as 100% of cell viability (% of cell viability = (sample O.D./control O.D.) × 100). Analyses were performed in three independent experiments, with four experimental replicates for each experimental point (% of cell viability = (sample absorbance/control absorbance) × 100).

### 2.8. Ki67 Immunofluorescence Analysis

Cells were grown on glass coverslips, washed twice with 1 mL of cold PBS, and fixed for 20 min in 3.7% paraformaldehyde in PBS and permeabilized with 0.3% Triton X-100 in PBS for 5 min. Cells were incubated in blocking buffer (5% FBS and 0.3% Triton X-100 in PBS) for 1 h at room temperature. Then, the cells were incubated overnight at 4 °C with ki67 antibody (Santa Cruz Biotechnology, Dallas, TX, USA) and successively for 1 h with the anti-mouse DyLightTM 488 secondary antibody (KPL, Gaithersburg, MD, USA) at room temperature. After staining of the nuclei with Hoechst 33242 dye (4′,6-diamidino-2-phenylindole; Life Technologies, Carlsbad, CA, USA), the cells were dried, mounted onto glass slides with ProLong Diamond AntifadeMountant (Thermo Fisher Scientific, Waltham, MA, USA), and examined with a confocal microscopy using a Nikon Eclipse TE2000-U (Nikon, Tokyo, Japan). A single composite image was obtained by superimposition of 6 optical sections for each sample observed. The collected images were analyzed by ImageJ software [39]. All the experiments have were repeated three times.

### 2.9. Statistical Analysis

Differences in the proliferation index for each experimental group, compared to the control one, were assessed using analysis of variance (ANOVA). To avoid bias due to the variability between the experiments, the factor defining the different experimental groups was crossed with a second factor defining the different experiments (two-way ANOVA). *p*-values lower than 0.05 were considered statistically significant. Figures are representative from all experiments that were realized during the study.

## 3. Results

### 3.1. Decreased Frequency of E. faecalis on Stool of CRC Patients

The presence of *E. faecalis* was investigated in the feces of healthy donors (HD) and CRC patients (CC) (Table 1) through two complementary approaches. Firstly, we performed the isolation of *E. faecalis* from fresh stools, and we revealed its presence in four over nine HD, while only in two over nine CC. In detail, we isolated 12 colonies from four healthy donors and four colonies from two CRC patients (Table 3). MALDI-TOF analysis showed that five of the total 16 isolated colonies were not properly classified, being identified as *E. faecium* or *E. galinarium* and were discarded. The remaining 11 colonies were correctly identified as *E. faecalis* and were subjected to clonal analysis. Following the RAPD-PCR assay, eight different clonal profiles were identified: Six (EFH01-EFH06) from the four healthy donors and two from the two (CRC01 and CRC02) CC patients (Table 3).

In addition, to evaluate the overall prevalence of *E. faecalis* in the same subjects, including non-culturable strains, we performed a q-PCR analysis using the total DNA extracted from ultra-freeze stool samples (maximum two months of storage at −80 °C). *E. faecalis*’ presence was detected in seven over nine healthy donors (HD01, HD02, HD03, HD05, HD06, HD07, HD09), and only in two over nine CRC patients (CC6 and CC8). Taken together, the isolation and q-PCR approaches allowed us to detect the presence of *E. faecalis* in the stools of 77% of healthy donors and in 22% of CRC patients.

### 3.2. Genotypic Characterization of E. faecalis’ Strains

The eight clonally different, representative, *E. faecalis* strains (Table 3) were genotyped by whole-genome-sequencing. First, we detected a different content of virulence genes (Appendix A
Appendix A). Then, the clonal analysis by determination of the sequence type (ST) revealed that selected strains were part of *E. faecalis* lineages previously found to be associated with: (i) Clinical isolates (EFH04/ST209, EFH06/ST16, CRC01/ST40, and CRC02/ST59); (ii) the healthy gut (EFH01 and EFH05 both ST21); and (iii) sporadically detected or not characterized clones (EFH02/ST47 and EFH03) (Table 3 and Figure 1). Subsequently, these eight strains were used to produce a pre-fermented RPMI medium.

### 3.3. Characterization of Secreted SCFAs 

Since LAB are known for their ability to produce SCFAs, we performed a quantitative analysis of a panel of SCFAs in pre-fermented medium samples obtained from all isolated *E. faecalis* strains (Table 4). As expected, acetic acid was the most significant SCFA produced by all strains, followed by iso-valeric acid. In addition, in all samples, we detected an SCFA not included in the panel of studied compounds. Taking into the account the retention time, distnace to iso-valeric acid, its fragmentation ions, and the relative abundances, we hypothesized the structure of 2-methyl-butyric acid. Since it is not included in the calibrated SCFAs, a quantitative evaluation was carried out using the calibration curve and reference ISTD of the iso-valeric acid. Data of the medium sample without fermentation of the studied strains did not show the presence of SCFAs, proving that they were produced by bacterial activities. Overall, we did not reveal substantial differences in the productions of SCFAs among all the isolated strains.

### 3.4. From the Isolated Strains, Three were Able to Decrease Tumor Cell Growth

MTT assays were made at 24, 48, and 72 h of HCT-8, HCT-116, SW-620, and CRL-1790 cells’ exposure to the pre-fermented medium with *E. faecalis* strains. In the preliminary experiments, we observed effects over the cell viability/proliferation after only 72 h, without differences regarding the control after 24 and 48 h (data not shown). Therefore, all the following experiments were performed at 72 h of pre-fermented medium exposure. The results revealed that secreted metabolites from the strains EFH01, EFH02, EFH03, and EFH04 (all from healthy donors) reduced the viability/proliferation of HCT-8 cells; similar results were also obtained with HCT-116 cells (except for EFH04), while no effect resulted in the SW-620 cell line (Figure 2). Secreted metabolites from EFH05, EFH06, CRC01, and CRC02 strains did not elicit any significant effects in the three CRC cell lines. Aiming to evaluate the potential effect of strains’ metabolites on normal cells, we used the embryo colonic cell line, CLR-1790. Noteworthy, we observed that *E. faecalis* strains’ metabolites did not significantly interfere with the viability/proliferation of these cells.

To better clarify the extent to which *E. faecalis* secreted metabolites affect cell proliferation, we employed Ki67 immunofluorescence staining as an accurate measurement of cellular proliferation (Figure 3). We observed that the Ki67 expression was decreased only on HCT-8 and HCT-116 cell lines cultured with the metabolites of the EFH01, EFH02, EFH03, and EFH04 strains, corroborating the results of the MTT assays (Figure 4). Finally, in agreement with the MTT assay, a complementary cell cycle phase distribution assay demonstrated a trend to decreasing G0/G1 and increasing S phases, mainly with the metabolites of the EFH03 strain (Appendix A).

## 4. Discussion

In this study, aiming to investigate the potential role of *E. faecalis* in CRC development, we compared the effect of *E. faecalis* strains isolated from CRC patients’ and healthy subjects’ stools on the viability/proliferation of different colorectal cells lines. Actually, gut microbes can regulate the colonic epithelial cells’ homeostasis (proliferation, differentiation, and apoptosis) and GM dysbiosis has been widely associated with CRC development [12,13,14]. In addition, metagenome-wide association studies on fecal samples have characterized numerous microbial markers of CRC [40,41]. Some authors documented a higher abundance of *E. faecalis* in CRC tumor and adjacent tissue [42], as well as in stool samples of CRCpatients [23,43], as compared to healthy subjects. However, in our cohort of patients, we did not confirm these previous findings. In detail, only two out of the nine CRC patients had cultivable *E. faecalis* in stool samples, while in healthy donors, we detected strains in seven out of the nine donors. In former studies [24,44], the authors used only a PCR-based experimental approach that did not unequivocally distinguish *E. faecalis* from other *Enterococci*. Conversely, we used a primer pair with a higher specificity to evaluate the presence of *E. faecalis*. So, a direct comparison among our data with previous ones is affected by this procedural bias. Although our observations were restricted to a limited number of subjects, the remarkable differences disclosed between CRC patients and healthy donors suggest that the majority of CRC patients possibly lose the physiological stool presence of *E. faecalis* that we supposed to have a protective role on gut health.

In support of our hypothesis, recent studies demonstrated that *E. faecalis* can affect the phosphorylation status of the peroxisome proliferator-activated receptor (PPAR)γ, triggering the activation of its downstream pathways [27]. PPARγ acts as a growth-limiting and pro-differentiating transcription factor in colonic epithelial cells [44] and contributes to innate antimicrobial immunity in the colon [45]. Deregulation of the PPARγ axis in CRC progression has been described [44] and efforts aimed to identify PPARγ agonists as anti-neoplastic agents are ongoing. Therefore, the disappearance of *E. faecalis* might reduce (or remove) the presence of a PPARγ modulator, thus contributing to a pro-tumoral milieu. It will be of interest to focus future studies on the definition of the precise phase of *E. faecalis* disappearance in the feces, during CRC progression.

Aiming to understand whether genetic differences documented by whole-genome-sequencing involve variation in the fermentation processes and their final products, we analyzed the concentrations of SCFAs on the culture supernatant of all isolated *E. faecalis* strains. SCFAs are physiologically active products, primarily obtained by the fermentation of soluble dietary fiber and resistant starch by commensal bacteria in the colon [46]. SCFAs can profoundly affect the inflammatory response [47,48], being closely linked to decreased CRC incidence [49], and also the adaptive immune response [50] (which has a key role in CRC progression, as we have previously demonstrated [16]). The in vitro assay demonstrated that SCFAs are able to inhibit the proliferation [51] of CRC cell lines by inducing their apoptosis [52]. The most frequent SCFAs in the colon and stool samples are acetic, propionic, and butyric acids, which are present in an approximate molar ratio of 60:20:20 [53,54].

SCFAs, especially those mentioned, are one of the most important products of the fermentation process and present diverse beneficial properties to human health, such as regulation of metabolism, inflammation, and disease (reviewed by Tan et al.) [55]. We did not observe significant differences in the SCFA concentrations among the isolated strains; however, the presence of 2-methyl-butyric acid, which was produced by all strains, was interesting, since this volatile organic compound can be used for discriminate analysis in the diagnosis of gastrointestinal diseases [56]. Therefore, its inclusion in the SCFAs’ panel for future studies could improve the correlation between the analytical data and biological evidence.

Bearing in mind the controversial role of *E. faecalis* in CRC development^6^ and the different behaviour of CRC cell lines in response to external stimuli (as well as each tumor having its own genotype, origin, and proliferative index [57]), we compared the effect of each isolated strain’s fermented products on the proliferation of three tumor cell lines: HCT-8 and HCT-116 and SW620s. As a result, we documented an antiproliferative effect of four (to note, all isolated from the healthy donors) of the eight isolated strains on HCT-8 and HCT-116 cells, while the others (two from CRC patients and two isolated from the HD02 subject) had no effect. In addition, we did not reveal any significant anti-proliferative effects in SW-620 cells, a very important result, on normal colonic CRL-1790 cells. Therefore, we disclosed a heterogeneous scenario, where only some *E. faecalis* strains derived from healthy donors possess inhibitory effects on primitive tumor-derived CRC cells’ viability/proliferation. Coherent with observation of a severe reduction of *E. faecalis* presence in the feces of CRC patients, the isolated strains did not show any anti-tumoral effect.

The differential ability of the isolated strains to impact colon cancer cells’ viability/proliferation, although perhaps not due to differences in SCFAs’ production, could depend on the production of numerous other metabolites (that we aim to investigate in future studies). For example, the GM products of various polyphenols’ metabolism (such as urolithins, benzoic acid, and 3-phenypropionic acid) are capable of inhibiting the proliferation of different human colon cancer cell lines [58,59]. In addition, isothiocyanates, the hydrolysis products of glucosinolates of various bacterial species, including *E. faecalis* [60], have been shown to have anti-carcinogenic properties in both in vivo and in vitro studies, causing cell cycle arrest and inducing apoptosis [61].

Overall, the obtained results lay the groundwork for a more in-depth characterization of the crosstalk between *E. faecalis* and CRC development, which is based not only on the evaluation of its presence in feces, but also looks at the biology of each different strain.

In conclusion, we demonstrated, for the first time, that some *E. faecalis* strains may have anti-tumoral properties and, in addition, different strains could have a highly heterogeneous content of virulence factors [62,63] (Appendix A). Therefore, in future investigations, we suggest the specific lineages of *E. faecalis* are investigate to clarify their role in both healthy hosts and in gut inflammation.

Finally, we suggest that *E. faecalis* strains with a probable protective role, such as EFH01, EFH02, and EFH03, should be investigated in order to be used as a potential probiotic treatment of pre-CRC intestinal disorders (e.g., inflammatory bowel disease (IBD) and adenoma) to prevent disease progression.

## Figures and Tables

**Figure 1 jcm-08-00388-f001:**
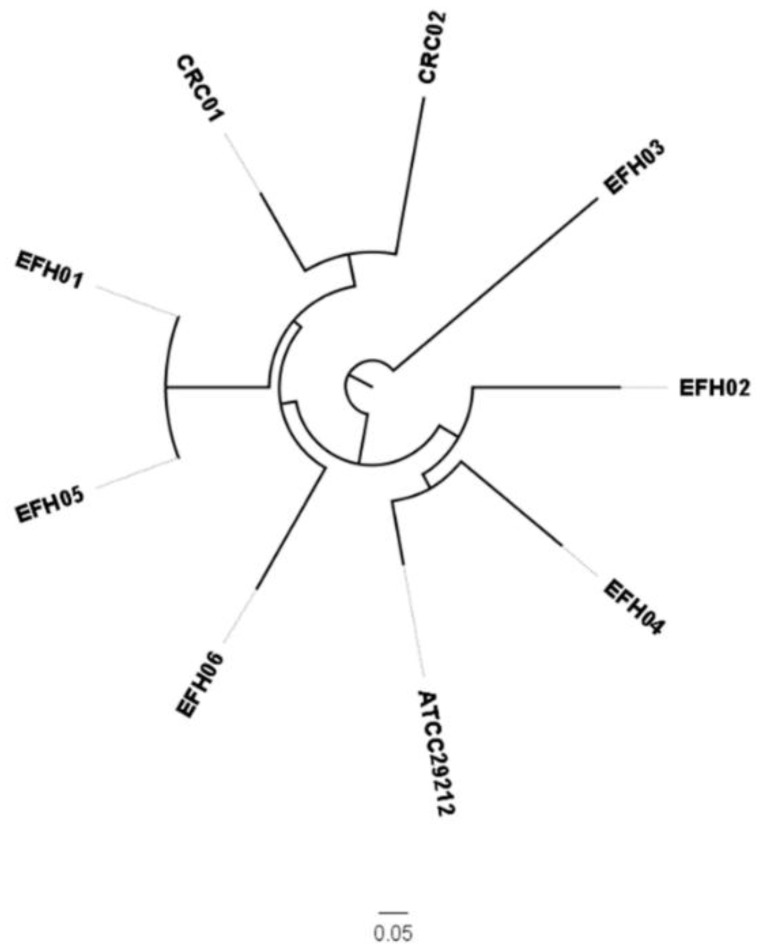
Phylogenetic tree of isolated *E. faecalis* strains. The phylogenetic tree was generated using the ATCC 29212 genome (GenBank acc. no. NZ_CP008816.1) as a reference. CRC: colorectal cancer; EHF: *Enterococcus faecalis* healthy.

**Figure 2 jcm-08-00388-f002:**
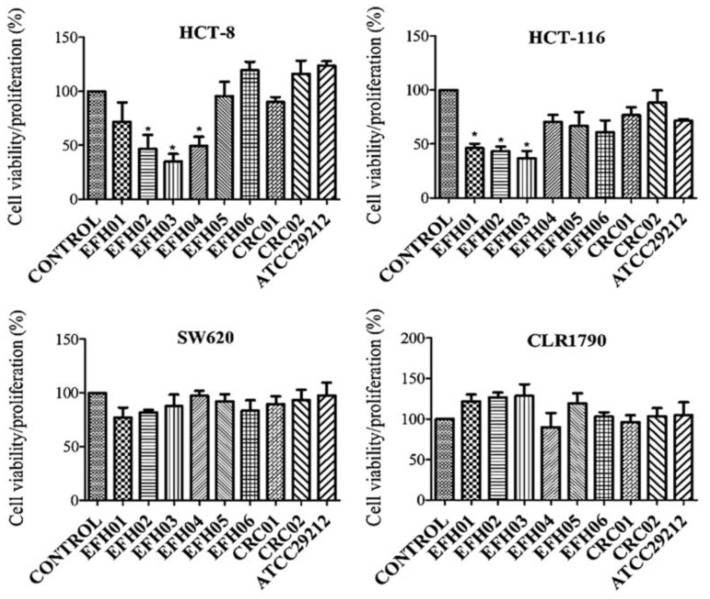
Estimated differences in the cell viability/proliferation index for each experimental group with respect to the control, one net of variability between the experiments after 72 h of exposure to *E. faecalis* metabolites. Error bars indicate the standard deviation (* *p* ≤ 0.05). HCT-8: Human ileocecal adenocarcinoma cell line; HCT-116: Human colon carcinoma cell line; SW-620:human colorectal adenocarcinoma cell line; CLR-1790: normal human diploid cell line

**Figure 3 jcm-08-00388-f003:**
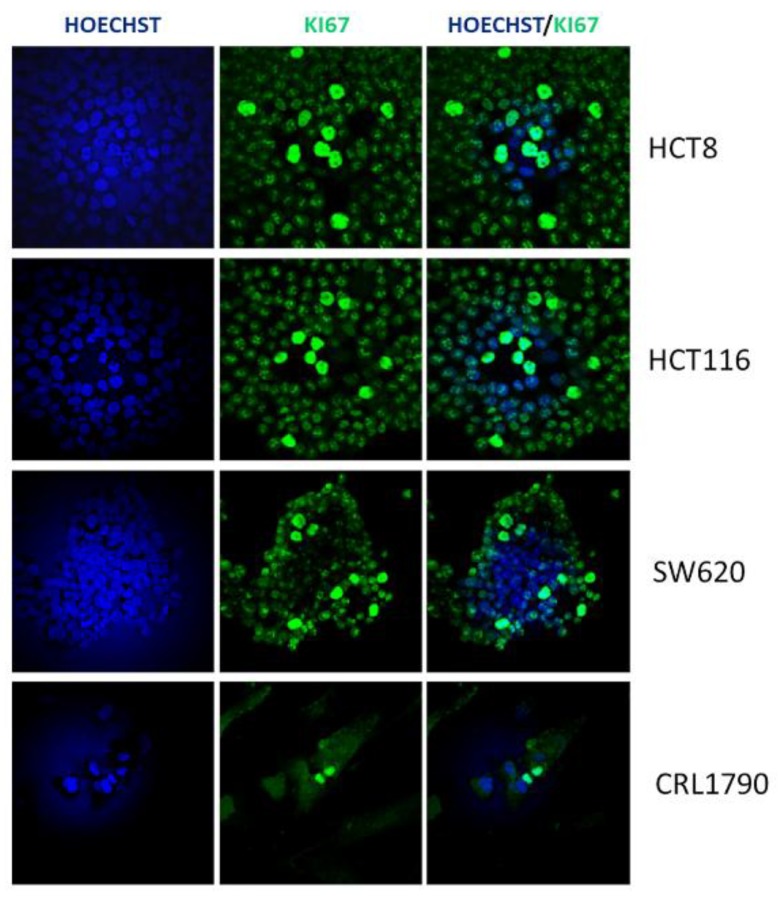
Representative double-labeled immunofluorescence images for Ki67 (green) and Hoechst (blue) in HCT8, HCT116, and SW620 CRC cells, and embryo colonic CLR1790 cells after 72 h of culture.

**Figure 4 jcm-08-00388-f004:**
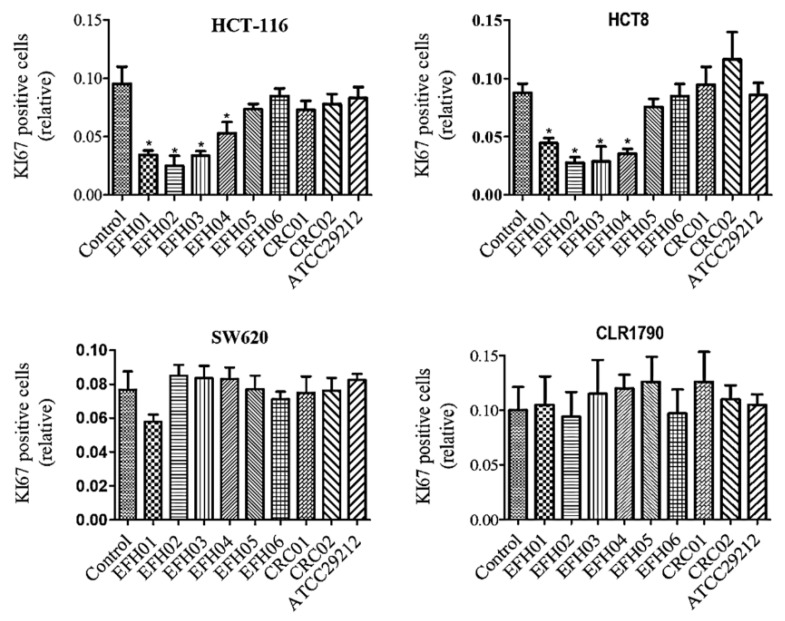
Ki67 positive HCT8, HCT116, SW620, and CLR1790 cells after 72 h of exposure to the metabolites produced from different strains of *E. faecalis*. Error bars indicate the standard deviation (* *p* ≤ 0.05).

**Table 1 jcm-08-00388-t001:** CRC patient and healthy donors’ characteristics.

Donors (Healthy—HD Colon Cancer—CC)	Age/Gender (♀/♂)	Histotype/Stage
**HD01**	57 ♀	n/a
**HD02**	60 ♂	n/a
**HD03**	56 ♂	n/a
**HD04**	53 ♀	n/a
**HD05**	68 ♀	n/a
**HD06**	65 ♂	n/a
**HD07**	68 ♀	n/a
**HD08**	48 ♂	n/a
**HD09**	52 ♂	n/a
**CC01**	78 ♂	Colorectal adenocarcinoma with moderate differentiation (pT3N0)
**CC02**	79 ♂	Colorectal adenocarcinoma with moderate differentiation (pT3N0)
**CC03**	68 ♀	Colorectal adenocarcinoma with moderate differentiation (pT2N0)
**CC04**	78 ♂	Colon intramucosal adenocarcinoma (pT2N0)
**CC05**	40 ♂	Colorectal adenocarcinoma with moderate differentiation (pT3aN1aMx)
**CC06**	78 ♂	Colorectal adenocarcinoma with moderate differentiation (pT1N0)
**CC07**	81 ♀	Colorectal adenocarcinoma with moderate differentiation (pT2N0Mx)
**CC08**	62 ♂	Colorectal adenocarcinoma with moderate differentiation (pT2N0Mx)
**CC09**	63 ♂	Colorectal adenocarcinoma with moderate differentiation (pT3N0Mx)

CRC: colorectal cancer; ♀: female; ♂: male; n/a: not available.

**Table 2 jcm-08-00388-t002:** Cell lines’ features and culture conditions.

Cell Line	Medium	Other Information
HCT-116	DMEM + 2 mM l-Glutamine + 100 U/mL Penicillin + 100 μg/mL Streptomycin + 10% FBS	(CCL-247™) ATCC^®^, Manassas, VA, USA. Colon; colorectal carcinoma, male, epithelial, primary tumor.
HCT-8	DMEM + 2 mM l-Glutamine + 100 U/mL Penicillin + 100 μg/mL Streptomycin + 10% FBS	(CCL-244™) ATCC^®^, Manassas, VA, USA. Colon; ileocecal colorectal adenocarcinoma; primary tumor, epithelial.
SW-620	DMEM + 2 mM l-Glutamine + 100 U/mL Penicillin + 100 μg/mL Streptomycin + 10% FBS	(CCL-227™) ATCC^®^, Manassas, VA, USA. Colon; derived from metastatic site: lymph node; Dukes’ type C, colorectal adenocarcinoma; male; epithelial
CLR-1790	50% DMEM + 50% Ham’s F12 Nutrient Mixture + 2 mM l-Glutamine + 100 U/mL Penicillin + 100 μg/mL Streptomycin + 10% FBS	(CCD 841 CoN) ATCC^®^, Manassas, VA, USA. Colon; normal; 21 weeks gestation fetus; epithelial.

HCT-8: Human ileocecal adenocarcinoma cell line; HCT-116: Human colon carcinoma cell line; SW-620: human colorectal adenocarcinoma cell line; CLR-1790: normal human diploid cell line; DMEM: Dulbecco’s Modified Eagle Medium; FBS: Fetal Bovine Serum.

**Table 3 jcm-08-00388-t003:** Isolated colonies of *E. faecalis* and characterized strains used for experimental protocols.

	Donors	*E. faecalis*/Total Enterococci	*E. faecalis* Strains Used	ST
**CRC patients**	CC1	-/-	-	
	CC2	-/-	-	
	CC3	-/-	-	
	CC4	-/-	-	
	CC5	-/-	-	
	CC6	2/3	CRC01	40
	CC7	-/-	-	
	CC8	2/2	CRC02	59
	CC9	-/-	-	
**Healthy donors**	HD1	3/8	EFH01	21
	HD2	4/10	EFH02	47
			EFH05	21 *
			EFH06	16
	HD3	3/8	EFH03	unk
	HD4	-/-	-	
	HD5	-/4	-	
	HD6	2/5	EFH04	209
	HD7	-/5	-	
	HD8	-/3	-	
	HD9	-/6	-	
**Total isolated strains**		16/54	8	

E. *faecalis*: *Enterococcus faecalis*; CC: Colon Cancer; HD: Healthy donors; EFH: *Enterococcus faecalis* healthy ST: sequence type; * EFH01 and EFH05 are both classified as ST21, even if they display genotypic and functional differences.

**Table 4 jcm-08-00388-t004:** Quantitative analyses of SCFAs in pre-fermented medium samples.

Samples	Acetic ± SD (ug/mL)	Propionic ± SD (ug/mL)	Butyric ± SD (ug/mL)	Iso-Butyric ± SD (ug/mL)	Iso-Valeric ± SD (ug/mL)	2-MethylButyric ± SD (ug/mL)	Valeric ± SD (ug/mL)
RPMI	n.q.	n.q.	n.q.	n.q.	n.q.	n.q.	n.q.
EFH01	393.5 ± 71.1	n.q.	n.q.	n.q.	4.7 ± 1.0	4.2 ± 0.8	n.q.
EFH02	268.7 ± 19.7	2.7 ± 2.5	4.2 ± 1.1	n.q.	3.3 ± 0.4	2.8 ± 0.2	0.3 ± 0.6
EFH03	379.7 ± 43.6	n.q	2.1 ± 1.8	n.q.	3.3 ± 0.5	3.3 ± 0.4	n.q.
EFH04	324.2 ± 23.9	n.q	1.0 ± 1.7	n.q.	4.8 ± 0.2	3.6 ± 0.2	n.q.
EFH05	394.1 ± 56.1	0.8 ± 1.4	1.0 ± 1.7	n.q.	4.6 ± 0.4	4.1 ± 0.4	n.q.
EFH06	381.2 ± 16.1	n.q.	n.q.	n.q.	6.4 ± 0.5	6.1 ± 0.6	n.q.
CRC01	417.5 ± 89.3	1.2 ± 2.1	2.6 ± 2.3	n.q.	4.2 ± 0.9	4.0 ± 0.8	n.q.
CRC02	399.7 ± 44.5	n.q.	3.3 ± 0.1	n.q.	3.7 ± 1.1	2.4 ± 0.4	n.q.
ATCC	380.6 ± 45.9	n.q.	n.q.	n.q.	4.7 ± 0.8	3.5 ± 0.6	n.q.

n.q.: below limit of detection of the quantitative method. Data are presented as mean ± SD (standard deviation). SCFAs: Short chain fatty acids; RPMI: Roswell Park Memorial Institute medium.

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
