# Peer review of "Differential Responses of Colorectal Cancer Cell Lines to Enterococcus faecalis’ Strains Isolated from Healthy Donors and Colorectal Cancer Patients"

_jcm, 2019, doi:10.3390/jcm8030388_

Reviewer 1 Report

1. expand the statistical section to include what parameters of central tendency were reported as for example see table 4 are those means +/- SD? clarify

2. in the results section provide clarification as to when the faecal samples were collected and how much time elapsed prior to DNA extractions...and what storage conditions were used for the samples if relevant.

3. Begin the discussion with what the study has found and not with more introductory sentences!

Author Response

Q1. Expand the statistical section to include what parameters of central tendency were reported as for example see table 4 are those means +/- SD? clarify

R1. As rightly suggested by the reviewer , we have clarified the statistical parameters reported in Table 4, Figure 2 and Figure 4.

Q2. In the results section provide clarification as to when the faecal samples were collected and how much time elapsed prior to DNA extractions...and what storage conditions were used for the samples if relevant.

R2. As suggested by the reviewer 1, we have included, in the Results, the details about the samples storage for E. faecalis isolation or total DNA extraction (please, see lines 289-300).

Q3. Begin the discussion with what the study has found and not with more introductory sentences!

R3. In agreement with the reviewer suggestions, we have changed and re-written the discussion beginning(please see lines 289-300).

Reviewer 2 Report

Reports in literature suggest that the metabolites produced by the GI microflora play an important role in both heath and disease.  In this research paper, the authors first isolated the Enterococcus species of bacteria from healthy and colorectal cancer (CRC) patients and strains were confirmed by genotyping analysis.  These bacteria were cultured and the medium containing the metabolites were studied for their effects on the proliferation of three different cancer cell lines (HCT-8, HCT-116, and SW620) as well as one normal cell line (CRL1790).  MTT assay and Ki67 immunofluorescence analysis were performed to determine the effect of the metabolites on cell proliferation.  They showed that secreted metabolites of E. faecalis  strains isolated from healthy donors decreased HCT-8 and HCT-116 cell proliferation, however it was not effective in SW620 and CRL1790 cells.  The metabolites of the strains isolated from CRC patients did not affect cell growth in CRC cell lines.  Their results show that only some E. faecalis strains derived from donors possessed inhibitory effect on cell proliferation of tumor-derived CRC cell lines.  They suggest that the heterogeneous effect observed on cancer cell proliferation due to the actions of the metabolites may be related to virulence factors associated with each strain.  They also suggested that strains such as EFH01, EFH02, and EFH03 could be better investigated as an additional probiotic treatment for some pre-CRC intestinal disorders.     

Minor comments:

1.      There are grammar mistakes throughout the body of the text.  It requires significant editing.  For example, line 26, instead of “cells lines”, please change to “cell lines”.  Line 26, the authors wrote that “We have firstly isolated and genotyping characterized the Enterococcus faecalis strains isolated”.  This sentence needs to be revised.    

2.       EFH01, EFH02, EFH03 and EFH04 appear to have significant effect on HCT-8 cell lines (MTT assay).  The authors can confirm these results by determining the effect of metabolites on cell number.  For this cells can be trypsinized with and without treatment with metabolites. 

3.      The authors state on line 242 “overall, we did not reveal substantial differences of SCFAs’ production among all the isolated strains”.  This suggests that the observed effect on cell lines is unlikely due to the secretion of SCFAs.  Please discuss what other metabolites may be responsible for the observed effects?   

4.      The authors can perform some additional experiments to determine how signaling pathways are affected by the metabolites on cell lines that are inhibited.  For example, does the phosphorylation status of the Peroxisome Proliferator-Activated Receptor (PPAR)Îł is changed? 

Author Response

 Q1.      There are grammar mistakes throughout the body of the text.  It requires significant editing.  For example, line 26, instead of “cells lines”, please change to “cell lines”.  Line 26, the authors wrote that “We have firstly isolated and genotyping characterized the Enterococcus faecalis strains isolated”.  This sentence needs to be revised.    

R1. We thank the reviewer for the remarks. We have checked and edited the entire manuscript in order to eliminate the mistakes, correcting also English style and grammar.

Q2.       EFH01, EFH02, EFH03 and EFH04 appear to have significant effect on HCT-8 cell lines (MTT assay).  The authors can confirm these results by determining the effect of metabolites on cell number.  For this cells can be trypsinized with and without treatment with metabolites. 

R2. Our data revealed that EFH01, EFH02, EFH03 and EFH04 affect viability/proliferation of HCT-8 and HCT-116 cells as determined by MTT assay. We agree with the reviewer that MTT/MTS/WST-1 analyses account for metabolic activity and not only for proliferation, thus our MTT analysis could not actually discriminate if the effect of metabolites is on cell proliferation or other metabolism activating processes, or both. For this reason, we performed KI67 scoring, which is a well-known accurate non-radioactive analysis of cellular proliferation, revealing the anti-proliferative activity of secreted metabolites. In the light of the above, we believe that our approach is a valid alternative to cell counting, and sufficient to corroborate our conclusions. Moreover, we edited the manuscript (please, see lines 255 to 275) to better clarify our approach and so, this point.

Q3.      The authors state on line 242 “overall, we did not reveal substantial differences of SCFAs’ production among all the isolated strains”.  This suggests that the observed effect on cell lines is unlikely due to the secretion of SCFAs.  Please discuss what other metabolites may be responsible for the observed effects?   

R3. We really thank the reviewer for the observation and the possibility to implement this concept in the discussion. We have discussed the role and the impact of other metabolites, involved in cancer cells viability/proliferation (please, see lines 352-359).

Q 4.      The authors can perform some additional experiments to determine how signaling pathways are affected by the metabolites on cell lines that are inhibited.  For example, does the phosphorylation status of the Peroxisome Proliferator-Activated Receptor (PPAR)Îł is changed? 

R4. We agree with the reviewer about the importance to investigate how, on cell lines that are inhibited; thesignaling pathways are affected by the metabolites. Unfortunately, we are not able to add additional experiment to date (also for the limited time of revision offered by the editors) but, as reported in the discussion section, we are now working on these investigations, which will be the subject of our future study

Round  2

Reviewer 2 Report

The authors have satisfactorily addressed the concerns raised by the reviewers. It is scientifically sound and the findings are highly are interesting.